

# Technical note: Image processing for continuous river turbidity monitoring - full scale tests and potential applications

Domenico Miglino[1], Khim Cathleen Saddi[1,5,6], Francesco Isgrò[4], Seifeddine Jomaa[2], Michael Rode[2,3], and Salvatore Manfreda[1]

[1] Dipartimento di Ingegneria Civile, Edile e Ambientale (DICEA), Università degli Studi di Napoli Federico II, Napoli, 80125, Italy

[2] Department of Aquatic Ecosystem Analysis and Management, Helmholtz Centre for Environmental Research - UFZ, Magdeburg, 39114, Germany

[3] Institute of Environmental Science and Geography, University of Potsdam, Potsdam-Golm, 14476, Germany

[4] Dipartimento di Ingegneria Elettrica e delle Tecnologie dell'Informazione (DIETI), Università degli Studi di Napoli Federico II, 80125, Italy

[5] Istituto Scuola Superiore Pavia (IUSS Pavia), Pavia, 27100, Italy

[6] Department of Civil Engineering and Architecture, Ateneo de Naga University, Naga, 4400, Philippines.

*Correspondence to*: Domenico Miglino (domenico.miglino@unina.it)

**Abstract.** The development of continuous river turbidity monitoring systems is essential, since it is a critical water quality metric linked to the presence of organic and inorganic suspended matter. Current monitoring practices are mainly limited by low spatial and temporal resolution, and costs. This results in the huge challenge to provide extensive and timely water quality monitoring at global scale. In this work, we propose an image analysis procedure for river turbidity assessment using different camera systems (i.e., fixed trap camera, camera on board of an Unmanned Aerial Vehicle, and a multispectral camera). We explored multiple types of camera installation setup during a river turbidity event artificially re-created on site. The outcomes prove that processed digital camera data can properly represent the turbidity trends. Specifically, the experimental activities revealed that single band values were the most reliable proxy for turbidity monitoring in short terms, better than band ratios and indexes. The best camera positioning, orientation and lens sensitivity, as well as daily and seasonal changes in lightning and river flow conditions, may affect the accuracy of the results. The reliability of this application will be tested under different hydrological and environmental conditions during our next field experiments. The final goal of the work is the implementation of this camera system to support existing monitoring techniques with early warning strategies and help in finding innovative solutions to water resources management.



**Graphical Abstract**

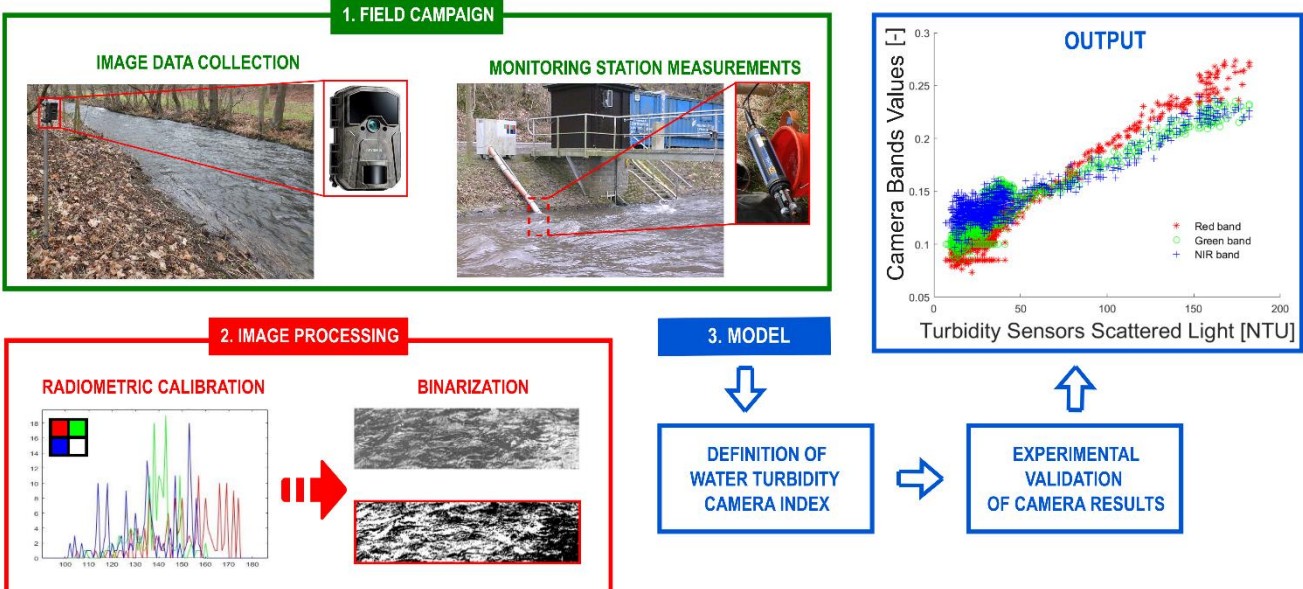

## 1 Introduction

Nowadays, compliance with the European Water Framework Directive and World Health Organization (WHO) guidelines for water quality is becoming more and more challenging (Santos et al., 2020; WHO chronicle, 2011), since human-related

activities and climate change are heavily impacting water resources. Therefore, fresh water will be a more and more valuable resource which deserves to be properly monitored, exploiting all available techniques, and also wisely managed (Manfreda et al., 2024). In this context, turbidity is a key factor for water quality monitoring and an optical property often used as an indicator of suspended particles and floating pollutants (Stutter et al. 2017; Tomperi et al., 2022). In inland water bodies, turbidity level and trophic state can strongly change with seasonality (Jalón-Rojas et al.,2015), soil erosion, extreme events, and farming

activities (Lu et al., 2023). Despite expensive costs for instruments and personnel, conventional in-situ monitoring techniques, using regular but not frequent sampling, return poor information to properly characterize temporal trends and spatial variability of hydrological and environmental conditions in river basins (Guo et al.,2020), usually underestimating the real loads (Gippel,1995).

In the last years, several innovations have been introduced in hydrological monitoring which exploit satellites, Unmanned

Aerial Vehicles (UAV) or camera systems in combination with image processing and machine learning techniques (Manfreda et al., 2018; Manfreda and Ben Dor, 2023). These methods offer the opportunity to provide large scale and detailed information on specific hydrological processes with relatively low costs. Within this context, many remote sensing applications for water quality applications have been developed (Ritchie et al., 2003; Ahmed et al., 2020), exploring the potential information coming





from the water spectral signatures (Gholizadeh et al. 2016) and investigating the dynamics of riverine ecosystems (Zhao et al., 2019; Lama et al., 2021).

Many of these studies developed turbidity estimation algorithms using satellite products, mainly for very large rivers, reservoirs (Potes et al., 2012; Constantin et al., 2016; Garg et al., 2020; Hossain et al., 2021) and coastal areas (Dogliotti et al., 2015). Unfortunately, satellite spatial resolution can't provide distributed estimations of water turbidity (WT) along the entire river network (Sagan et al., 2020) and the frequency of data collection is limited by satellite revisit period, usually of 5-10 days for Sentinel 2 and Landsat 8 (Jia et al., 2024). Recent studies are starting to investigate the perspective of digital cameras and low-cost optical sensors for river turbidity monitoring (Gao et al., 2022; Droujko & Molnar, 2022). However, no studies focused yet on the potential of image analysis applied in a real riverine environment. Such an application could definitively grant continuous high-frequency data, across the inland water bodies even without spatial resolution issues. Moreover, latest advances in computer vision techniques can certainly help us in extracting water quality information from images. The present study explores the use of an image-based monitoring procedure for river turbidity estimation. It was carried out within a real river where an artificial perturbation of the water turbidity has been used to find the optimal configuration for camera systems, the best performing band and range of applicability of the procedure. The manuscript contains a short introduction with the background used, then the field experiment and methods are illustrated in Section 2, finally results are discussed providing our final remarks.

## 1.1 RGB image acquisition and interpretation

The use of digital cameras in river monitoring activities can increase our knowledge of the real status of water bodies, solving the above-mentioned cost and data resolution problems of the existing techniques. The challenge of image-based procedures is the proper red, green, and blue (RGB) signal interpretation and processing. Goddijn et al. (2009) affirm that cameras can be seen as three-band radiometers, able to measure the water-leaving spectral response. The actual water upwelling light $R_w$ that reaches the camera lens, schematically shown in Figure 1, is the sum of various reflectance components of the suspended particles ($R_s$), the riverbed background ($R_b$) and the water itself. One component could prevail over the others, depending on the variability of hydrological (water level, flow velocity, etc.) and environmental (suspended solids concentration, floating pollutants, etc.) characteristics of the river. Digital cameras receive these inputs and return a signal in terms of RGB pixel intensity values.



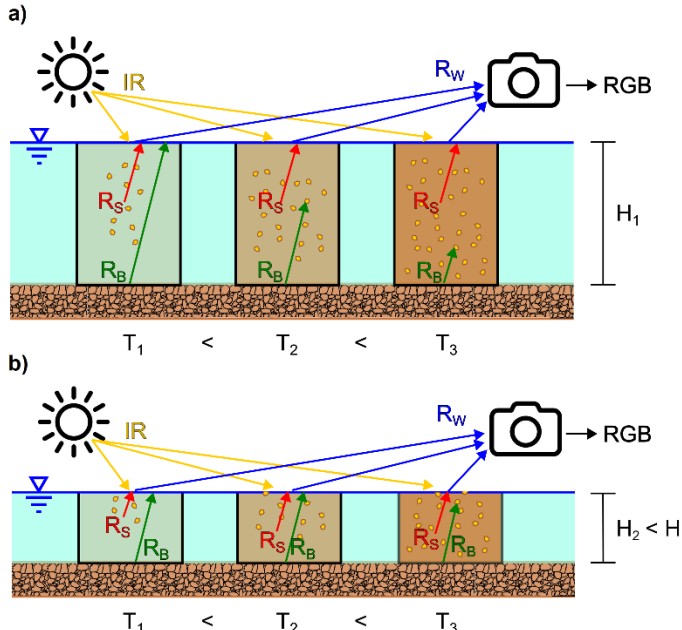

**Figure 1: Light behaviour within shallow (b) and non-shallow water (a): The solar irradiance (IR) passes through the water, whose reflectance (Rw) is influenced by the background (Rb) and suspended particles (Rs) presence, by varying the water level (H). Finally, the total water reflectance (Rw) signal is caught by the digital camera that produces an image with different pixel intensities of red, green and blue values (RGB).**

## 1.2 Turbidity image-based measurements

In the literature, there is a robust relationship between digital camera output and water quality indicators. Each of these methods requires specific solutions to provide trustable results based on the absolute water colour estimation under changing light conditions. For instance, Goddijn and White (2006) fixed a pipe around the camera lens to avoid external reflections for adjusting the image data collection. Leeuw and Boss (2018) developed an innovative smartphone app called "Hydrocolor", using images of the sky and a grey card nearby the camera's view field as radiometric references for turbidity estimation from the pictures of the water. Nevertheless, the reliability of their results strongly depends on the quality of the unsupervised image data coming from the citizens and the environmental conditions, resulting in inaccurate estimates for water surface roughness and changing weather conditions. More recently, Ghorbani et al. (2020) provided a continuous monitoring camera tool for suspended sediment concentrations (SSC) and turbidity, by using image analytics methods and machine learning techniques. They found evidence of correlation between SSC and camera images in their experiments under laboratory-controlled conditions.

In real riverine environments there are many more variables to consider. The image reflectance can be strongly influenced by several factors regarding river flow and light conditions variability. Moreover, different bands could provide several information about the water status, both considering single bands and their combinations. Nechad et al. (2010) demonstrated that single bands in the red and NIR (Near-Infrared) spectral ranges give a robust outcome in mapping total suspended matter



in coastal turbid waters using several satellite data sources. However, the choice of single bands or their combination is dictated by the concentration of the suspended solids and the type of floating pollutants, as well as water depth and riverbed background. In addition, the accuracy of the estimates is certainly influenced by camera positioning and orientation with respect to the examined river section. Our analysis involved the installation of a trap camera (TC), a multispectral camera (MSC), and an

Unmanned Aerial Vehicle (UAV) in order to examine the best spectral response of red, blue, green, NIR bands and their combinations, as well as the best camera installation setup.

The purpose of the field campaign was to conduct tests on the potential practical applications of the image processing for river turbidity monitoring. This tool can promote the development of early warning networks at river basin scale, moving water research forward thanks to a large increase of data on water bodies and the reduction of operating expenses.

## 2 Materials and methods

### 2.1 Full-scale experiment

The field experiment took place in February 2023, in the monitoring station of Meisdorf, Germany, better described in Miglino et al. (2022). The selected river section was the Selke River, within the TERENO (TERrestrial ENvironmental Observatories) global change exploratory catchment managed by the Helmholtz Association, Germany (Wollschläger et al., 2017). Recently,

the Bode basin has gone through prolonged droughts from 2015 to 2019, resulting in changes in land use, water quantity and quality. This could potentially impact also the suspended solids load and pollutants concentration. In this experiment a synthetic turbidity event was recreated by adding kaolin clay into the water, upstream enough from the monitored river cross-section, to ensure the complete mixing of the tracer (Figure 2). Kaolin is usually exploited to prepare turbidity standard solutions. In addition, it is a harmless, easy to handle, and a cheap mineral, which is also a common silicate in natural soils

and sediments.

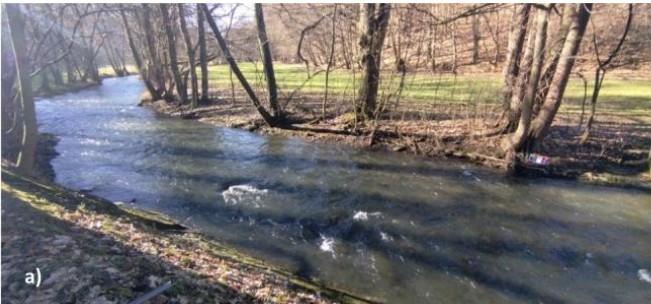 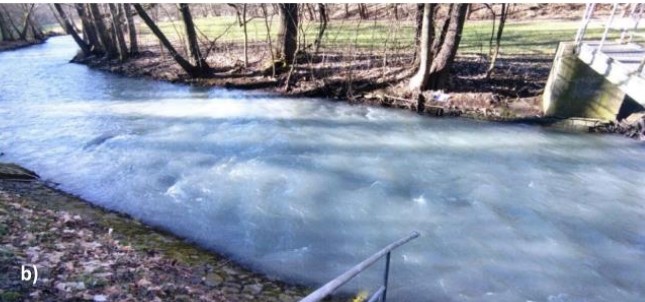

**Figure 2: View of Selke River, before (a) and during (b) the synthetic turbidity peak in the field experiment.**

We conducted the tracer experiment on 14 February 2023, adding 50 kg of kaolin tracer evenly distributed to the whole stream cross section at 12:05, 700 m upstream of the monitored river section. The mean flow velocity was 0.47 m/sec, the flow



discharge was 2.3 m3/s, while the water level was 0.54 m, and the width of the river section was 9 m. The turbidity level

started to artificially increase after 12:20, the peak was reached around 12:30, and the event ended after 13:00.

Several monitoring instruments were used during the experiment: three low-cost trap cameras (TC-Ceyomur CY50), one

multispectral camera (MSC-Tetracam ADC Snap) and one unmanned aerial vehicle (UAV-DJI Mavic 2 Enterprise Dual) were

placed in different positions. By looking at Figure 3, the MSC, in red, was installed on the bridge and the square in red

represents its field of view. Two TCs were installed on the left riverbank (LRB) and the right riverbank (RRB), while the third

TC was fixed on the bridge. The last camera was on the UAV, which ensured a zenithal field of view on the river section,

indicated by the light blue square. The flight height of the UAV was 5 m.

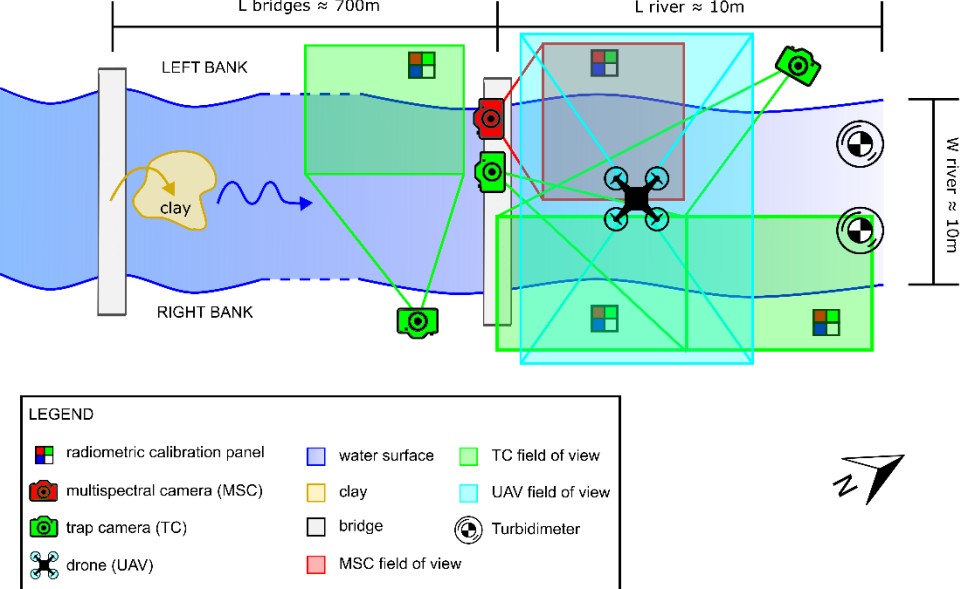

**Figure 3: Plan of the monitored river section during the field experiment, showing the generated synthetic turbidity events, using**
**kaolin clay tracer.**

The collected data could be affected by sun-glint, shadows and other external light sources. For these reasons, it's essential to

find the optimum camera installation design, for minimizing the uncertainties from water images. The camera data was

compared to the measurements of the turbidimeters installed underwater in the river cross-section (Figure 4). They were located

with a distance of 2 m each from the right and left stream bank for ensuring to detect the complete mixing of the suspended

solids.



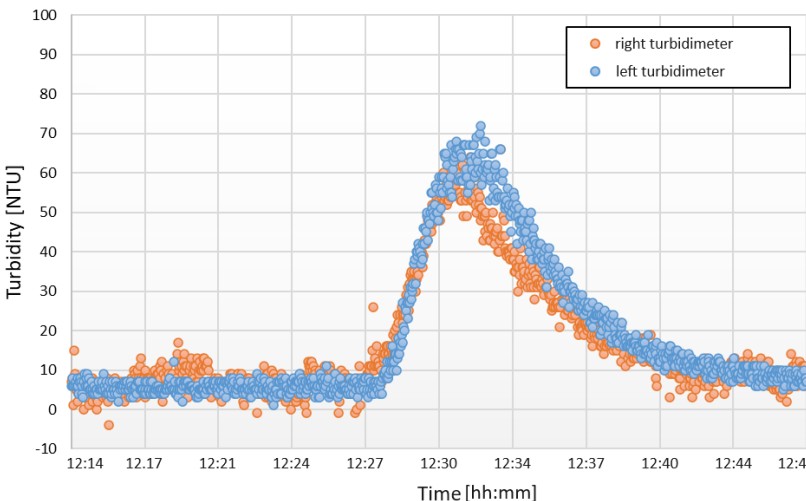

**Figure 4: Optical measurements of turbidity, using two turbidimeters installed on the right and the left riverbed sides during the experiment.**

The frame set in Figure 5 displays all the camera's fields of view along the stream during the experiment. The region of interest

(ROI) was selected making sure that it included only the water surface area. The mean of the pixel values inside the ROI area was considered as the representative value for the turbidity level for each single picture. Moreover, a radiometric calibration panel (RCP) was installed within the picture area, close to the investigated water surface. It consisted of a waterproof plastic laminated panel containing the reference RGB colour values for the image processing steps below.

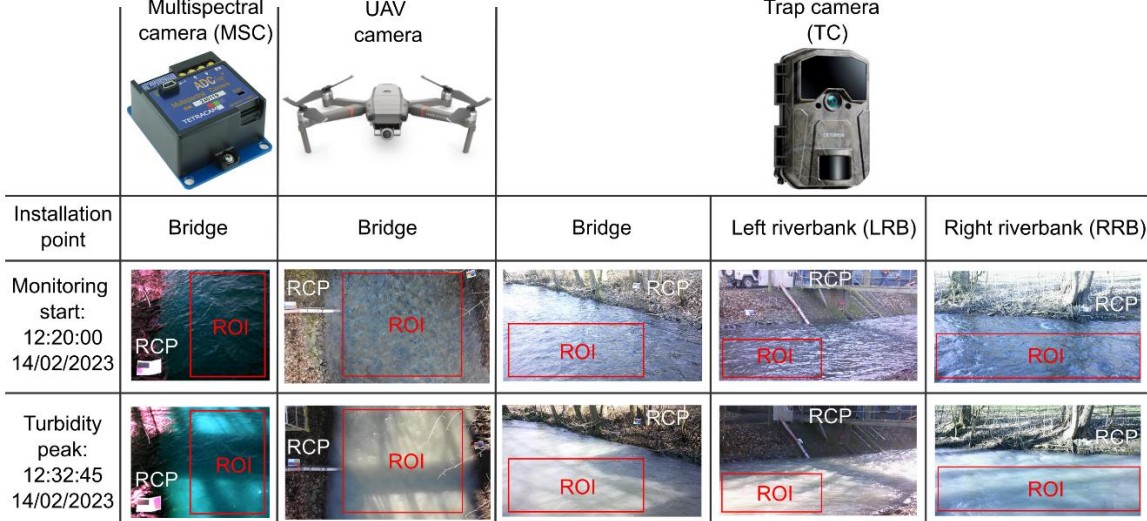

**Figure 5: Frame set of the camera types, fields of view and installation points during the start and the peak event time of the field experiment.**



## 2.2 Image processing procedure

This work defines and tests an image-based method that takes into account variables occurring within time- and site-specific riverine environments. It's important to build a robust procedure, since the acquired camera data cannot provide information
as they are because they are not yet compared to physically meaningful units. Herein, our workflow (Figure 6) sets out the general algorithm of image processing for WT analysis, starting from a correct image extraction and stabilization. Then, the steps of radiometric calibration and binarization allowed us to homogenize and select the relevant features from the image data. Finally, the WT indexes were defined from processed signals and validated by field measurements to train the model and quantify the river turbidity level.

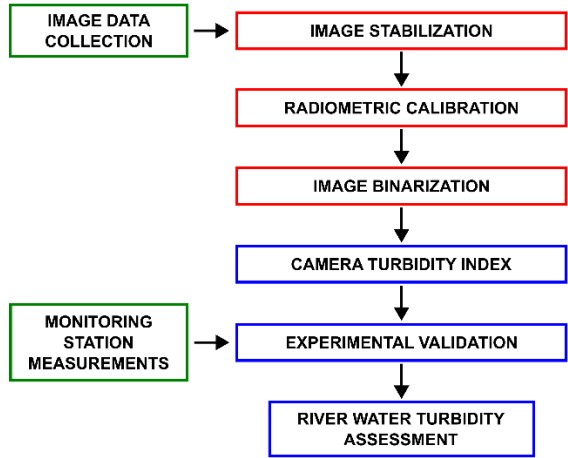


**Figure 6: Image processing procedure workflow.**

### 2.2.1 Image extraction and stabilization

The image data were stored as timelapses with a set frame rate depending on the type of camera. The number and the format of the extracted images were fixed for each timelapse frame, also collecting frequencies, to ensure the correct comparison
between cameras and measurement data.

The image sequences were involved in the image stabilization process, because the position of the objects in the scene could be shifted by environmental factors and camera instability. The image stabilization techniques performed an automatic detection and matching of features within the selected image area close to the RCP (Figure 7). In particular, we used the Harris-Stephens corner detection algorithm to identify feature points and remove apparent movements and jitter within the field of
view in the videos (Harris & Stephens, 1988; Abdullah et al., 2012). This step was necessary to grant the correct detection of the RCP and ROI areas location required for the following image analysis processes.



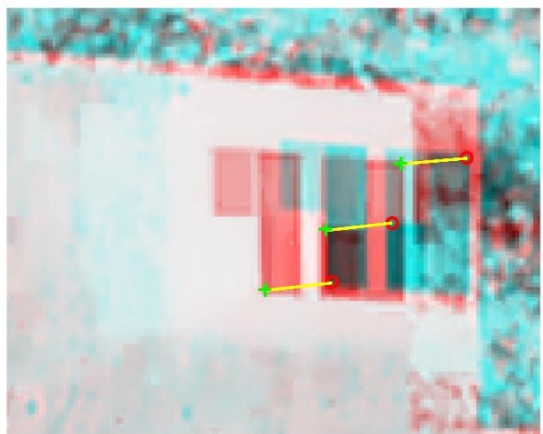

**Figure 7: Stabilization of the RCP coordinates for each extracted frame. The yellow lines show the panels shift during a camera monitoring period of two months.**

### 2.2.2 Radiometric calibration

For absolute radiometric correction in monitoring activities where the distance between the ground and the camera is less than 100 meters, it is assumed that the atmosphere is not influencing the light signal. Nevertheless, other site-specific and meteorological variables can still affect the camera measurement (Daniels et al., 2023). The radiometric signal of an object is influenced by the geometry of the measurements, depending on the relative positions of the sun, the measured object and the optical sensor. The direct-diffuse ratio, the atmosphere absorption, and scattering of the solar radiation in the space from the object to the sensor, and also the camera sensitivity are all significant factors influencing the natural illumination conditions. Using reference targets or recognized radiometric standards within the scene is necessary to convert the uncalibrated image pixel intensity (PI) values, also called digital numbers (DNs), into radiometrically meaningful units such as reflectance or radiance (Guo et al., 2019; Kinch et al., 2020). In this experiment, we chose a simplified design of the radiometric calibration panel with inside the assumed reference values ($RV_{rc}$) of the maximum PIs of red, green and blue for all the image (Figure 8.b), considered as the mean of the respective single band values of the pixels inside the panel squares.



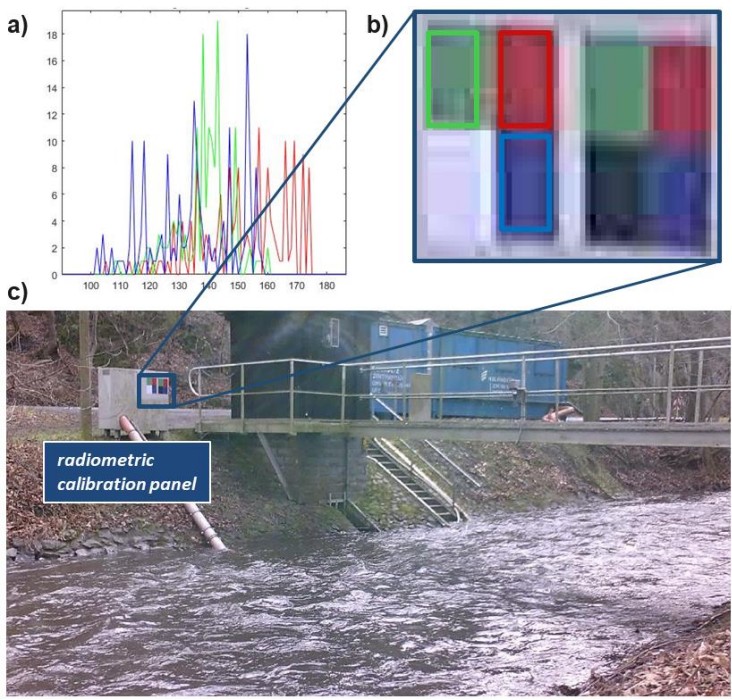

**Figure 8: Example of radiometric calibration procedure applied to a TC image, using the reference mean RGB values (a) of the radiometric calibration panel (b) installed within the camera field of view (c).**

The image PI values were reassigned considering the RCP reference values frame by frame, for each band, as follow:

$$PIrc = \{1, PI/RV_{rc}\}, \tag{1}$$

where $RV_{rc}$ is the panel reference value of red, green and blue for the radiometric calibration process.

PI values range for each band goes from 0 to 255, but we considered normalized values between 0 and 1. Once the radiometric

signal is correctly calibrated, the effect of changes in light conditions on the image information is substantially reduced. Therefore, some image areas could still be affected by sun glare and overly intense shadow. These pixels must be removed by binarization because they do not contain useful information about the water reflectance.

### 2.2.3 Image binarization

Image binarization techniques convert images into binary representations typically applying predetermined thresholds to

grayscale or RGB values. Here, the adopted procedure follows the Otsu's approach (Otsu, 1975). The global threshold was defined, separately for each band and for each frame, as a result of the minimization of the weighted variance of two clusters. All the values above this threshold are replaced with 1, while the other values with 0. The procedure involves iterations through every image pixel and counts the occurrence of each intensity. From the combination of the binarized single bands (Figure 9)





it is possible to remove the pixel with signal distortions due to the effect of sun glare and the shadows. In this way, only the
actual water reflectance information can be retrieved from the pictures.

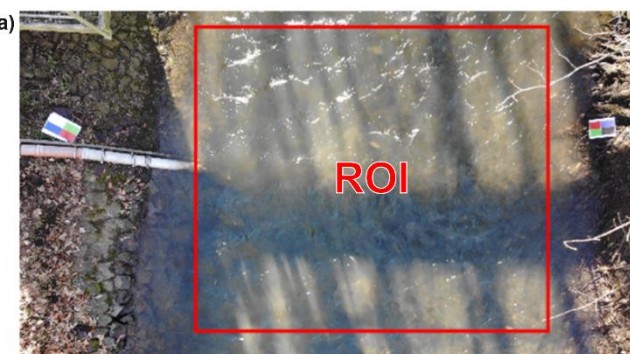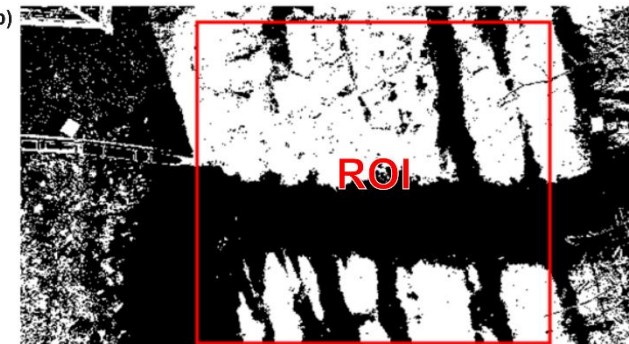

**Figure 9: UAV image before (a) and after (b) the binarization procedure.**

**2.2.4 Water turbidity camera index**

All the information coming from the camera bands could be properly considered for getting information on WT level in the
monitored river site. We selected the most representative remote sensing applications for each band and index, as explained in
Table 1.

| Band Combination | | Camera | Reference |
|---|---|---|---|
| Red / Green | | TC, MSC, UAV | Wang et al.,(2006) |
| NDTI | | TC, MSC, UAV | Lacaux et al.,(2007) |
| NDWI | | MSC | McFeeters,(1996) |
| Single Bands | Red | TC, MSC, UAV | Leeuw & Boss,(2018) |
| | Green | TC, MSC, UAV | Khorram et al.,(1991) |
| | NIR | MSC | Zhu et al.,(2020) |

**Table 1: Camera band combinations selected for on-site river turbidity remote measurements**

Single green and red bands were considered the most representative signals for identifying the turbidity. The NIR band is also
effective in detecting very high turbidity levels. Some ratios between these bands were taken into account too. We selected
from literature the ratio between red and green, the normalized difference turbidity index (NDTI) derived from RGB-imagery
and the normalized difference water index (NDWI) that combine NIR and green bands.



## 3 Results

Our experiments highlighted the capability of digital cameras to detect variations in WT level. We observed that digital camera results are influenced by several factors, such as the type of sensor adopted, the camera sensitivity, position and orientation. In particular, MSC results in Figure 10 describe distinct behaviours of the single bands. Red and green bands can capture turbidity increases above the measured value of 20 NTU, while the NIR band spectral response is much lower. Since the NIR band is totally absorbed by the water surface, its substantial changes can be detected only for very high concentrations of 220 suspended particles. If we consider the NDWI, the correlation between the camera and turbidimeter data becomes more consistent, as well as those of red and green bands.

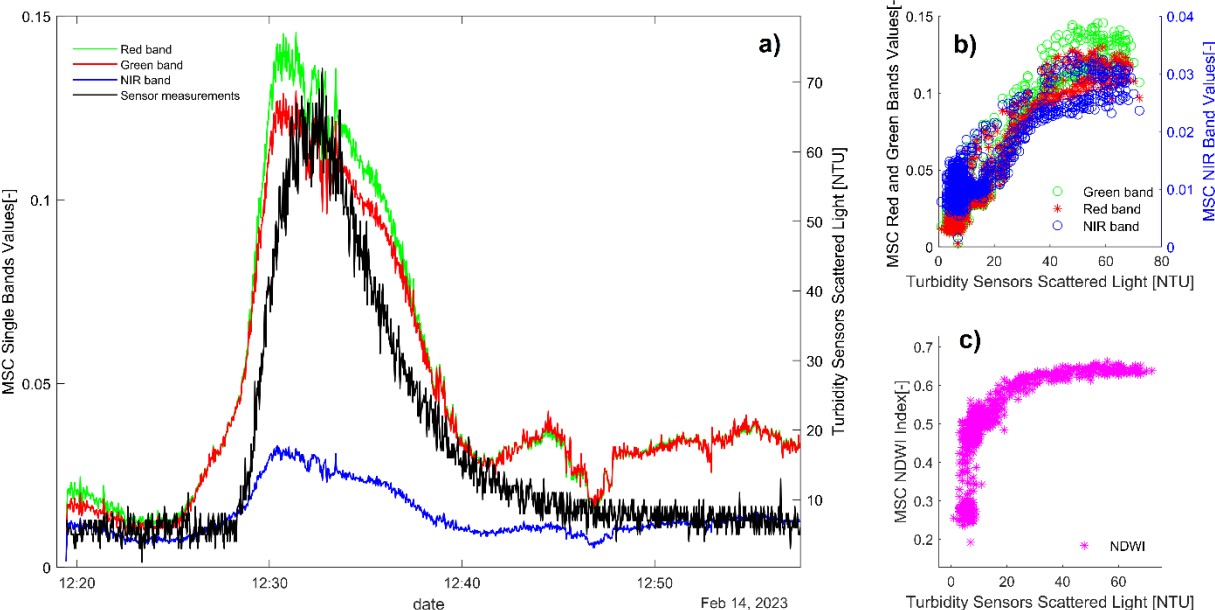

**Figure 10: a) Comparison between turbidimeter measurements and data from the multispectral camera (MSC) installed on the bridge; b) Scatter plot of the MSC single bands values and on-site measurements; c) Scatter plot of the MSC NDWI values and on-**
**site measurements.**

Trap camera outcomes are reported in Figure 11, where it is possible to observe different patterns depending on the installation position. Red and green band signals, from the TCs installed on the LRB and RRB, follow the measurement curve for the entire monitoring period, while the TC installed on the bridge seems to be influenced by the variation of the light condition during the first part of the experiment, before exceeding turbidity values higher than 50 NTU. Moreover, all the TCs results 230 show the same intensity values in correspondence to the turbidity peak, except those from LRB matching the measures but with lower PI signals. The most reliable TC bands ratios and indexes were those from the RRB position (Table 2).



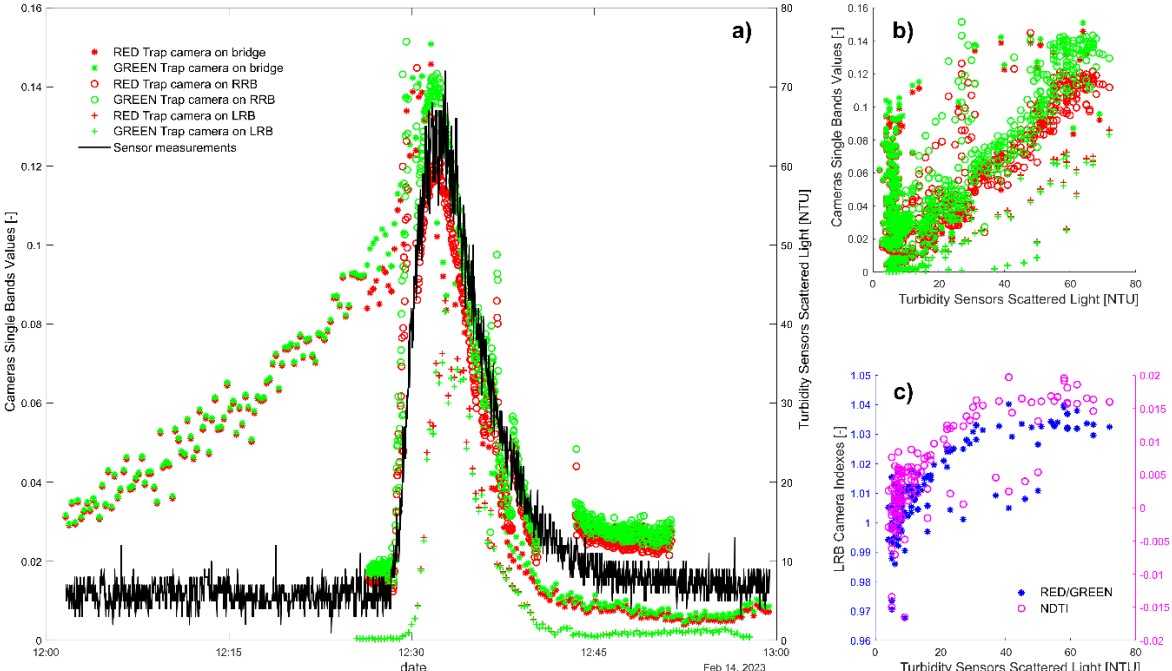

**Figure 11: Comparison between turbidimeter measurements, red and green band values from all the trap cameras installed on site; b) Scatter plot of all the cameras green band signal (in green) and red band signal (in red); c) Scatter plot of the NDTI (in magenta) and Red/Green (in blue) indexes from the trap camera installed on the left riverbank (LRB).**

The UAV camera returned partial data (Figure 12), due to the loss of signal that caused a break in the recordings for 5 minutes immediately after reaching the measured turbidity peak. Therefore, there is a good correspondence between UAV bands signal and the turbidimeter data. In the second part of the recording, we can observe that the bands signals are lower than those during the peak, but they don't exactly fit the measurements decreasing curve. That's because the white balance setting of the camera was on and this resulted in a discrepancy in the intensity of the starting signals, but not in terms of variations.





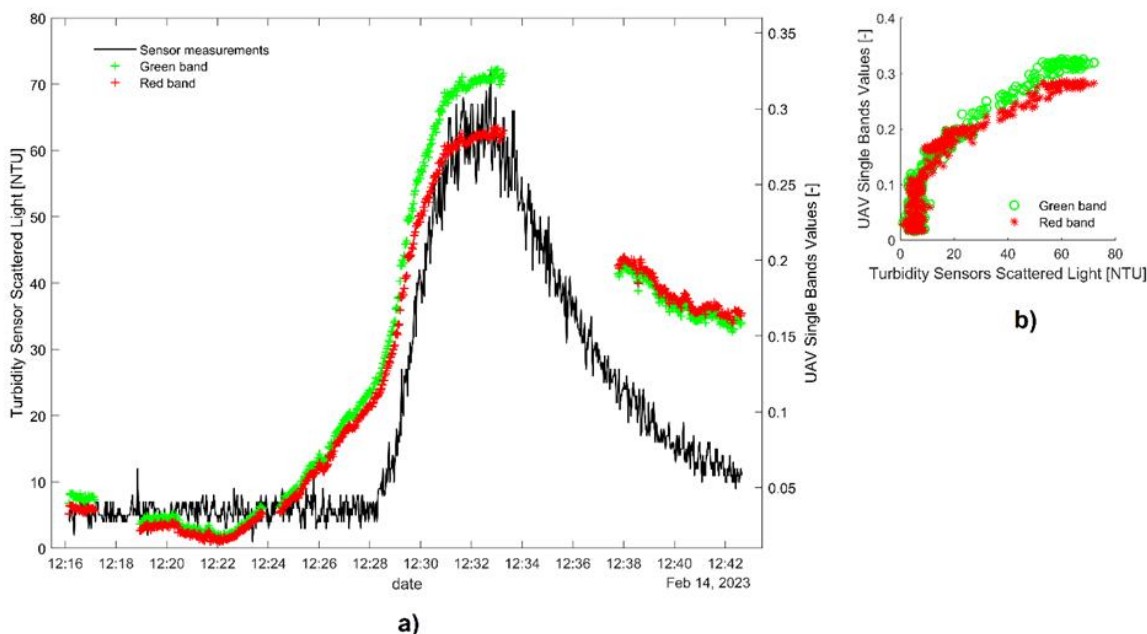

**Figure 12: a) Comparison between turbidimeter measurements and data from the UAV camera; b) Scatter plot of the variables.**

## 3.1 Performance metrics

Table 2 summarizes the performance in terms of linear and quadratic R-squared correlation coefficients, considering the
different camera types, bands, view angles, and installation setups selected for the field tests.

|  |  | BRIDGE | | | | ZENITHAL VIEW | | LEFT BANK | | RIGHT BANK | |
|  |  | M.S. CAMERA | | TRAP CAMERA | | DRONE | | TRAP CAMERA | | TRAP CAMERA | |
|  |  | LIN.$R^2$ | QUAD.$R^2$ | LIN.$R^2$ | QUAD.$R^2$ | LIN.$R^2$ | QUAD.$R^2$ | LIN.$R^2$ | QUAD.$R^2$ | LIN.$R^2$ | QUAD.$R^2$ |
|---|---|---|---|---|---|---|---|---|---|---|---|
| SINGLE BANDS | RED BAND | 0.89 | 0.91 | 0.25 | 0.31 | 0.76 | 0.88 | 0.82 | 0.85 | 0.89 | 0.89 |
| SINGLE BANDS | GREEN BAND | 0.91 | 0.92 | 0.27 | 0.32 | 0.83 | 0.90 | 0.82 | 0.85 | 0.88 | 0.88 |
| BAND RATIOS | RED/GREEN | 0.12 | 0.16 | 0.01 | 0.06 | 0.01 | 0.34 | 0.59 | 0.64 | 0.37 | 0.38 |
| INDEXES | NDTI | 0.11 | 0.15 | 0.01 | 0.07 | 0.01 | 0.34 | 0.58 | 0.63 | 0.38 | 0.39 |
| INDEXES | NDWI | 0.52 | 0.65 |  |  |  |  |  |  |  |  |

**Table 2: Linear and quadratic correlation coefficients R² of the cameras bands compared to turbidity measurements, considering different camera types and installation points selected for the experiment.**

Red and green single bands can describe turbidity variations better than band ratios and indexes, for all the cameras used in
this short-term experiment. Moreover, the MSC installation allowed us to understand the potential uses of bands beyond the
visible spectrum. NIR band seems to have good performance for high concentration of suspended particles that reflect a
consistent part of its radiation. In fact, considering combined MSC bands, the best performance comes from the NDWI index
that involves the NIR and green bands.



## 4 Discussion

Regarding the experimental setup, the presence of the submerged turbidimeters in both river sides (Figure 4) ensured to quantify the horizontal variability of the turbidity level along the river cross section. The vertical variability of turbidity on the water column is not so significant for a river as small as Selke, with a registered maximum water level of 1 meter. However, the proposed image-based procedure shall also apply for bigger rivers, since cameras capture the light from the entire water column until a mid-high turbidity level is reached. Once achieved this threshold (Figure 2.b), only the water surface can be
investigated by the camera.

Interesting results were observed for all three RGB bands, since we used a white clay to increase the turbidity level. Further experiments with multiple tracers as inputs, changing the colours and particle concentrations will help to gauge the effectiveness of the procedure. What we expect from a generalized application of the procedure, on the light of this and the past field tests experiences (Miglino et al., 2022), is the greater reliability of the red band, for variable suspended particles
characteristics, and better performances of band ratios and indexes, for long term monitoring in different hydrological conditions.

### 4.1 Range of applications

The comparison of all experimental data in Figure 13.a shows that the initial band signal responses were different for each camera, depending on lens sensitivity, position, field of view, angles from water surface and orientation with respect to the
sun's apparent motion axis. One way to homogenize these results is to consider the increment of band signals, starting from a band signal for clear water condition, referred to the image frame with minimum measured NTU value. Figure 13.b shows the increments between the red band signals (Red) for each frame and for the clear water conditions ($Red_0$). Since the UAV data was split into two distinct videos, some of the primary turbidity event occurred during the pause between the two recordings, hence it was excluded. In addition, the UAV camera was the only one with a pre-set automatic light balance, making it
impossible to integrate these data with the others.




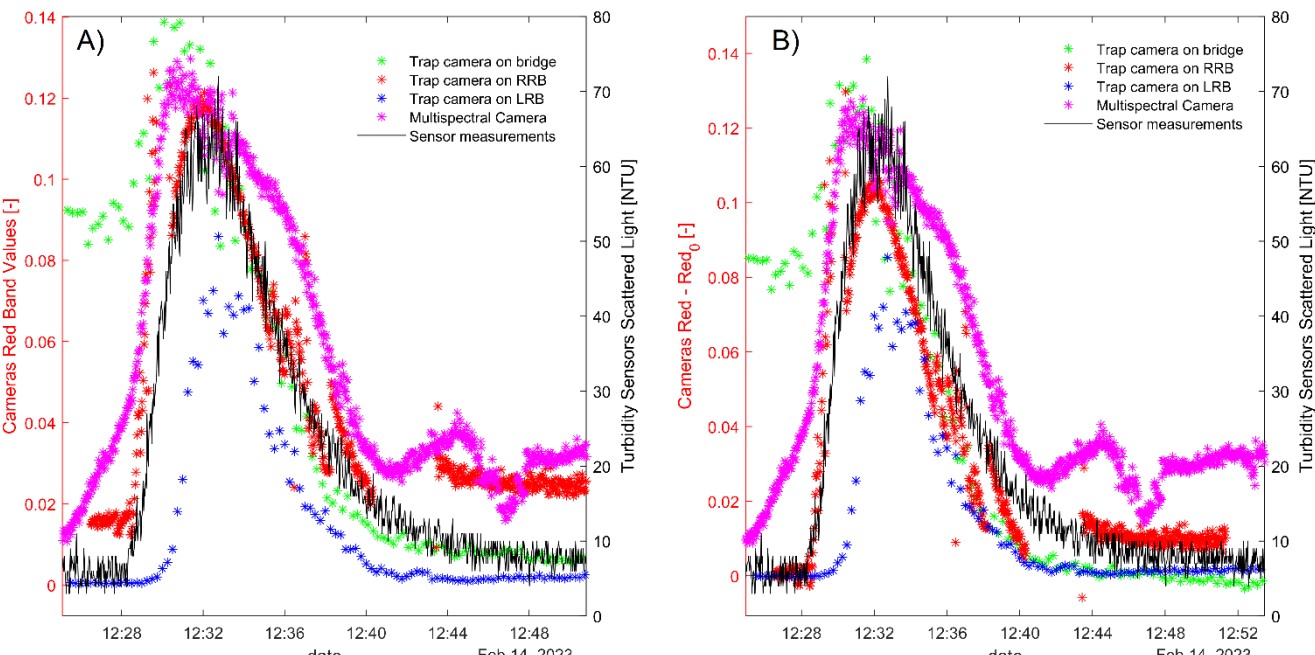

**Figure 13: Comparison of turbidimeter measurements and all the cameras' data, in terms of red band values (A) and increments (B) starting from a clean water condition ($Red_0$).**

It is worth to observe how there is a better overlap for the Red- $Red_0$ increment curves than the single red band signal curves. In fact, the identification of the PI for the clear water could remove the mismatches between the initial PIs detected by the camera, both due to changes in light and camera lens sensitivity. Next experiments will face this issue, especially for shallow water conditions, where the visibility of the riverbed background could become a reference value of water clarity, regardless of site- and time- specific variabilities.

## 4.2 Implications in river monitoring practices

Prior to this work, image analysis for water turbidity was predominantly conducted using satellite data for large rivers or through the use of camera data and specific optical sensors in laboratory. The added value of our study lies in the development of a monitoring procedure that can be directly implemented on-site. This allowed us to test the method under real conditions and to optimize the camera installation for future applications in various environments.

The proposed image processing procedure offers significant advances in river monitoring practices by providing a near real-time, continuous, and automated system for water turbidity assessment. This approach can complement current monitoring practices, addressing their limitations in data availability and resolution, especially for small or inaccessible rivers where existing methods are impracticable. Furthermore, the use of remote sensing technology minimizes environmental disturbance, aligning with sustainable monitoring practices. The widespread of this procedure could significantly increase the amount of





available information on water status at basin scale, thereby enhancing our understanding of the ecohydrological dynamics involved in river processes.

## 5 Conclusions

305 The experimental activities revealed that single band values were the most reliable proxy for turbidity monitoring in short-term observations, better than band ratios and indexes. The opposite could be true for long term observations, since single band signal tends to be more influenced by the variability of light and flow conditions. The advantages of using this procedure are multiple. Field tests proved that cameras, even the cheap models, can produce reliable turbidity estimates continuously in time. Moreover, they can be easily installed, in greater numbers than turbidimeters without the burden of cost, along the river network, providing a comprehensive knowledge of the river basin status. On-site tests are still on-going. It will allow us to 310 acquire a significant set of data, covering many environmental and hydrological conditions, to fully understand how to optimize the characteristics of the camera and the installation setups.

The practical application of this image-based procedure could create an innovative early warning network, not only limited to turbidity, but also provide a great potential for other water quality (e.g. chlorophyll-a) and water related (e.g. macroplastic) monitoring applications (Manfreda et al., 2024), advancing and supporting the existing river monitoring techniques. The next 315 natural step is the involvement of these water quality estimation algorithms in a citizen science approach. Within this context, our research group is developing a smartphone app for river monitoring (https://sites.google.com/view/riverwatch/home-page?authuser=0), focusing in particular on macroplastics and turbidity, that are the most easy-to-capture water quality information collectable by the people. The selected case study for the research activity is the Sarno River in Italy, one of the most polluted rivers in Europe. The real implementation of a continuous image-based river monitoring network like this, can 320 offer new options to water resources management strategies and the preservation of aquatic ecosystems.

### Data availability

The experiment data can be provided by the corresponding author upon request.

### Author contribution

DM and SM did the conceptualization; DM, SM, SJ and MR planned the field experiment and performed the measurements; 325 DM, SM, SJ and MR analyzed the data; DM and SM wrote the manuscript draft; SM, MR, FI, SJ and KCS reviewed and edited the manuscript.



**Acknowledgements**

The present research has been carried out within the projects "OurMED: Sustainable water storage and distribution in the Mediterranean", which is part of the PRIMA Programme supported by the European Union's Horizon 2020 Research and Innovation Programme under Grant Agreement No. 2222; the PRIN project "RiverWatch: A Citizen-Science Approach to River Pollution Monitoring" (project number 2022MMBA8X, CUP: J53D23002260006) funded by the Italian Ministry of University and Research, and RETURN Extended Partnership and received funding from the European Union NextGeneration EU (National Recovery and Resilience Plan - NRRP, Mission 4, Component 2, Investment 1.3 - D.D. 1243 2/8/2022, PE0000005).

**Financial support**

This work was supported by NextGenerationEU, the Italian Ministry of University and Research (MUR) and the Horizon 2020 Research and Innovation Programme.

**Appendix A: Acronyms and scientific units**

LRB - Left riverbank

MSC - Multispectral camera

NTU - Nephelometric turbidity unit

RCP - Radiometric calibration panel

RGB – Red, green and blue: colour representation model used on the digital screen

ROI - Region of interest

RRB - Right riverbank

TC - Trap camera

UAV - Unmanned Aerial Vehicle

WT - Water turbidity

DN - Digital number [-]

PI - Pixel intensity [-]

IR - Solar irradiance [$W/m^2$]

NDTI - Normalize difference turbidity index [-]

NDWI - Normalize difference water index [-]

NIR - Near-infrared radiation band [0.78–3 μm]

$R_b$ - Reflectance of the riverbed background [-]

$Red_0$ - Red band value for clear water condition [-]



RV$_{rc}$ - RGB bands reference value of the radiometric calibration panel [-]

R$_s$ - Reflectance of the suspended particles [-]

R$_w$ - Reflectance of the water [-]

SSC - Suspended sediment concentration [gr/l]

**Competing Interests**

The contact author has declared that none of the authors has any competing interests.

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
