# Peer review of "Technical note: Image processing for continuous river turbidity monitoring - full scale tests and potential applications"

_EGUsphere, 2024_

## Referee Comment (RC2)

Title: Image processing for continuous river turbidity monitoring - full scale tests and potential applications

The paper is well written but need following questions to be answered.

1. You should mention more numerical results in your abstract.
   The Introduction section is quite dense with information. So it makes harder to the reader to get the key points. Consider simplifying the language and structure to improve readability.
2. Please ensure that the main objectives and contributions of the study are clearly stated.
3. Consider adding more detailed literature review could strengthen the justification for the study.
4. Clearly outline what specific research questions the study aims to answer and how the findings will contribute to existing knowledge in the field.
5. In the Methodology section, consider adding a brief summary at the beginning of the section that outlines how each method contributes to the overall goal of estimating turbidity.
6. Ensure consistency in the use of terminology throughout the section.
7. Discuss any limitations or uncertainties in the results.
8. Consider discussing the implications of the findings in terms of MONITORING.
9. Summarize the key findings and their implications in a concise manner.
10. In conclusion, provide a brief overview of the main objectives and methodology of the study before diving into the conclusions.
11. Based on the findings, suggest recommendations for future research or practical applications.

---

## Author Comment (AC1)

[Figure]

**Figure 1R: Comparison of turbidimeter measurements (in black) and trap camera NDTI index (in red) during long-term monitoring from February to April 2022**

[Figure]

**Figure 2R: Comparison of normalized turbidimeter measurements and cameras red band values during the short-term experiment of February 2023**

[Figure]

**Figure 3R: Plots of results precision (A, D) and accuracy (E, F) for each camera (B), considering the entire turbidity event and only the peak time (C, D, E, F) during the short-term experiment of February 2023**

[Figure]

**Figure 4R: Comparison of turbidimeter measurements (in black) and number of analyzed ROI pixels in MSC data (in red) during the short-term experiment of February 2023**

[Figure]

**Figure 5R: Comparison of turbidimeter measurements (in black) and the standard deviation of ROI pixels intensity in multispectral camera red band (in red), during the short-term experiment of February 2023**

[Figure]

**Figure 6R: Comparison of turbidimeter measurements (in black) and number of analyzed ROI pixels in LRB trap camera data (in red) during the short-term experiment of February 2023**

[Figure]

**Figure 7R: Comparison of turbidimeter measurements (in black) and the standard deviation of ROI pixels intensity in LRB trap camera red band (in red), during the short-term experiment of February 2023**

---

## Author Response (AR1)

Dear Reviewers and Editor,

We would like to thank the first reviewer for the constructive suggestions that allowed us to improve the quality of the manuscript and clarify some concepts. We carefully considered all your observations and reviewed the manuscript accordingly.
We'd also like to thank the second reviewer for the reasonable questions. We answered them all, in order to make our research work more reliable and comprehensive.
In the following, specific comments by Reviewer 1 (RC1) and by Reviewer 2 (RC2) are addressed by the Authors' comments (AC) using the same order adopted by the reviewers. Within the marked-up manuscript version, all the changes have been commented with the corresponding acronyms of reviewer comments "RC1" and "RC2".

**Author's Response to Reviewer 1**

**RC1.1**: This study reveals very interesting first results that seem promising, but I think that there are not enough experiments yet to indeed prove the suitability. For the justification of a published manuscript more thorough experiments and their analysis are needed, e.g., using the two turbidimeter measurements of a longer period (i.e., not just the Kaolin experiment) and analyse the fit of the camera data. I would suggest to maybe consider to publish the data and first findings in form of a data publication. The outside setup is great to reveal the many challenges inherent in mapping turbidity of rivers, but I still believe more tests at different sites and/or also laboratory conditions can heighten the relevance of this study a lot. Why was especially this site chosen?

**AC1.1**: We fully agree with the reviewer on the importance of conducting further experiments. In Figure 1R, our preliminary long-term monitoring data of 2022 is shown. The installation setup was very simple: only one trap camera installed beside the river, a radiometric calibration panel installed on the other side, and a turbidimeter measurement each 15 min(see Figure2 of Miglino et al., 2022. The trap camera data were compared with the sensor measures. The camera data during the night and poor light conditions were not considered. We can already observe a clear correspondence between the variables, especially for the two main turbidity events of February, but also conflicting results for the other low-moderate turbidity peaks.

[Figure]

**Figure 1R: Comparison of turbidimeter measurements ("x" symbols in black) and trap camera NDTI index ("*" symbols in red) during long-term monitoring from February to April 2022**

On the light of these results, we felt the need to investigate the factors influencing the monitored process by designing the short-term experiment. We can include these considerations in the work, for better clarify why we needed to carry out the proposed study.

The study we propose here is characterized by a wide range of camera types and installation setups, making it ideal for the understanding and analysis of the reliability of all the key variables of the process in real world settings.

In our opinion, we provide comprehensive results for the purposes of this study, that are the description and the field testing of our simplified image processing procedure tailored for water turbidity estimation in real riverine environments, exploring its potential in actual river monitoring practices.

Moreover, we are currently finalizing the results of a second experiment, where the procedure has been tested under different hydrological conditions, using different cameras and tracers with different characteristics of the suspended particles. This will help us to generalize the method as much as possible.

For these reasons, we also think that literature on laboratory tests on this topic already exists and we want to focus on a procedure directly implemented on-site.

As mentioned in line 107, the Meisdorf monitoring station's river section is thoroughly detailed in our previous work (Miglino et al., 2022)

This chosen river belongs to the Bode basin, that is one of the meteorologically and hydrologically best-instrumented catchments in Central Germany by Helmholtz Centre for Environmental Research (UFZ), with a long-term time series on water quantity and quality. This allowed us to use the monitoring station measurements as a validation of our experimental results. Additionally, the choice of this river section was guided by these criteria: visibility of the river cross-section, variability of the parameter to be monitored

(turbidity in our case), convenience and durability of the installation points of the cameras. Several UFZ monitoring stations have been visited along the Bode River and Meisdorf Station proved to be the most suitable site.

Reference:
*Miglino D, Jomaa S, Rode M, Isgro F, Manfreda S. Monitoring Water Turbidity Using Remote Sensing Techniques. Environmental Sciences Proceedings. 2022; 21(1):63. https://doi.org/10.3390/environsciproc2022021063.*

**RC1.2**: The authors use Kaolin as turbid river load. This material has a very bright color. Thus, the influence of differently colored sediment would be necessary to indeed assess the suitability of the introduced approach, which is especially the case that at different sites different sediment colors are likely due to different geological source regions. In general, how representative is Kaolin for this site as turbidity?

**AC1.2**: Kaolin was an excellent candidate for the field experiment, since it is commonly exploited to prepare turbidity solutions. In addition, it is harmless, easy to handle, and a cheap mineral, which is also a common silicate in natural soils and sediments. Moreover, Kaolin is also used for reproducing high turbidity wastewater and simulate suspended sediment loads in laboratories.
Nevertheless, we agree with the reviewer, but we preferred not to add too many variables in this first study, aimed only at providing the conceptualization of the procedure and its feasibility in a real riverine environment.

References:
*Isoyama, R., Taie, M., Kageyama, T., Miura, M., Maeda, A., Mori, A., & Lee, S. S. (2017). A feasibility study on the simultaneous sensing of turbidity and chlorophyll concentration using a simple optical measurement method. Micromachines, 8(4), 112.*
*Ma, J., Fu, X., Xia, W., Fu, K., & Liao, Y. (2019). Flocculation of a high-turbidity kaolin suspension using hydrophobic modified quaternary ammonium salt polyacrylamide. Processes, 7(2), 108.*
*Schieber, J., Li, Z., Yawar, Z., Cao, X., Ashley, T., & Wilson, R. (2023). Kaolinite deposition from moving suspensions: The roles of flocculation, salinity, suspended sediment concentration and flow velocity/bed shear. Sedimentology, 70(1), 121-144.*
*Radomirovic, T., Smith, P., & Jones, F. (2013). Using absorbance as a measure of turbidity in highly caustic solutions. International Journal of Mineral Processing, 118, 59-64.*

**RC1.3**: In regard of the trail cameras, the authors do not seem to consider that these cameras use the Bayer pattern, i.e., the pixel values RGB area already interpolated from the originally captured raw data. How does that influence the usage of the bands red and green as absolute measures for turbidity, which are however not the pure color channels anymore? This needs to be discussed more and also related to the multi-spectral camera, which indeed measures individual ("pure") bands.

**AC1.3**: Actually, we have considered that issue, but our intention is to identify a simplified image processing procedure, affordable and easy to apply in many practical contexts, not

only using expensive multi- and hyper-spectral optical sensors, but also common RGB cameras.

For the curiosity of the reader and the exhaustiveness of the study, we can add these following considerations to the Discussion section:

"To enable a proper comparison of all the variables, we decided to normalize them, as shown in Figure 2R, since the data came from several sources, such as turbidity sensors and camera bands. The min-max normalization was chosen as the most suitable technique for our analysis because it scales the variables to a common range from 0 to 1, enabling direct comparison of their trends despite differences in units and ranges. This helps to highlight correlations between the turbidity sensor readings and camera outputs during the turbidity event, preserving the shape of the original distributions and maintaining relative relationships between data points.

[Figure]

**Figure 2R: Comparison of normalized turbidimeter measurements and cameras red band values during the short-term experiment of February 2023. The colour schemes used in this figure are accessible to persons with colour vision deficiencies.**

Within the Figure 2R, the mean of all the normalized camera bands values is also reported in red. The normalization made it easier to assess camera performances in relation to the measurements of turbidity. Additionally, it enables a consistent comparison among different camera types, avoiding the issue of the Bayer pattern, used in digital cameras, that provides us RGB values already interpolated, unlike the multispectral camera that measures pure bands.

[Figure]

**Figure 3R: Plots of results precision (A, D) and accuracy (E, F) for each camera (B), considering the entire turbidity event and only the peak time (C, D, E, F) during the short-term experiment of February 2023. The colour schemes used in this figure are accessible to persons with colour vision deficiencies.**

Figure 3R describes the performance during the synthetic turbidity event of February 2023, focusing on precision and accuracy of the normalized variables. The precision is meant as the absolute difference between the turbidimeter measures and the camera data, while the accuracy as this difference divided by the measures, which represents the percentage of the error. In particular, Figure 3R.A shows the precision of the mean of the normalized camera bands. The red dots in this picture show the lower discrepancy between the measures and the image data, pointing out higher reliability of the camera data in reflecting the turbidity levels, during the peak time, from 12:29 to 12:39, for the actual increase of turbidity (NTU>20). Figure 3R.B describes the normalized red band values for the five different cameras over the entire turbidity event. The results show that most cameras exhibit a relatively similar means and range of values, except for the LRB TC presenting lower values but still aligned with the measures. Figures 3R.C, D and E highlight again the less range of variability and the best matching of the camera data with the turbidity sensor readings, during the peak time rather than the entire event. Figure 14F illustrates the behaviour of normalized red band values of the different camera types during the turbidity peak time. It shows notable variability between the cameras. On one hand, MSC results display the largest spread, indicating lower accuracy than TCs. On the other hand, MSC box plot, together with LRB TC, is the only one without outliers and point standing on the whiskers, explaining his strong correlation with measurements reported in Table 2.
Overall, the results from both the short- and long-term data suggest us that the camera lens sensibility is not the only factor to consider. Camera orientation, installation setups, bands available, and also the intensity of the measured event can influence the monitoring performance."

**RC1.4**: Chapter 1.2 (and Table 1): The introduced methods for turbidity measurement using images should be described in a bit more detail. In the current form, it is not possible to

understand how these methods work without consulting the papers. At least some high-level overview should be given.

**AC1.4**: We already gave a comprehensive overview in that section on the existing studies and projects related to this topic, it is not a review paper.
Moreover, as explained in section 4.2, there aren't significative and specific examples in literature on this. The proposed research work wants to be a novel and added value to river monitoring.

**RC1.5**: Within the RoI, the mean pixel value is considered as proxy for the turbidity. However, besides this value it would also be of interest to consider the standard deviation of the pixel values, e.g., to assess the influence of speckle and shadows and to get a general idea of the variation of the digital numbers.

**AC1.5**: Only the mean of the processed pixels within the RoI is considered as a proxy for turbidity. The image processing procedure allowed us to minimize the misinformation from the images, reducing the effect of changing light conditions with radiometric calibration, and removing saturated pixels or in strongly shaded areas with binarization.
The following Figures 4R, 5R, 6R, and 7R describe the variability of the standard deviation and the number of analyzed RoI pixels, for both MSC and TC camera.

[Figure]

**Figure 4R: Comparison of turbidimeter measurements (line in black) and number of analyzed ROI pixels in MSC data ("*" symbols in red) during the short-term experiment of February 2023**

[Figure]

**Figure 7R: Comparison of turbidimeter measurements (line in black) and the standard deviation of ROI pixels intensity in multispectral camera red band filtered ("o" symbols in red) and unfiltered ("*" symbols in red) data, during the short-term experiment of February 2023**

[Figure]

f

**Figure 6R: Comparison of turbidimeter measurements (line in black) and number of analyzed ROI pixels in LRB trap camera data ("*" symbols in red) during the short-term experiment of February 2023**

[Figure]

**Figure 7R: Comparison of turbidimeter measurements (line in black) and the standard deviation of ROI pixels intensity in LRB trap camera red band filtered ("o"symbols in red) and unfiltered ("*"symbols in red) data, during the short-term experiment of February 2023**

It is worth to observe that the considerations in the previous answer are confirmed by these figures: the effectiveness of the procedure increase for high turbidity level.

The number of RoI pixels that can be analyzed is directly proportional to the turbidity, both for MSC (Figure 4R) and trap camera (Figure 6R). These results show that the resolution of the camera data must be considered to ensure a minimum number of analyzable pixels, especially for less sensitive camera, such as the trap camera.

In Figures 5R and 7R, it is clear that the filtering of both the camera data by the image processing procedure, significantly reduces the standard devations of the ROI pixels intensity, in particular for low-moderate turbidity conditions, where the spatial variability of the ROI pixel intensities is particulary high for the unfiltered data.

**RC1.6**: The authors consider an RGB-panel for calibration. The idea is interesting as it would enable the usage of a low-cost and simple approach. However, some further information in regard of the influence of the viewing angle and the sun angle would be of interest. Is there some calibration needed?

**AC1.6**: For this experiment, we decided to investigate these influences, changing positions and viewing angles of the different cameras. From both the short- and long-term results, we observed that the more the RGB panel area and the RoI area are under the same sun exposure, the stronger the effect of the radiometric correction, especially for long-term monitoring. A preliminary removal of the camera frames with RGB panel and RoI showing strong differences in sun exposure, is needed, or, if the panels are more than one, it is possible to select the panel closer to RoI conditions.

**RC1.7**: The binarization process of the image is not very clear to me. Based on what is the decision for the threshold made. In Figure 9 it seems that there are some problems with the approach. Shadow regions are removed, but the panel at the river side is partly in the shadow. Thus, what are valid values then. Why would the shadow region not be considered if the reference is in the shadow? What is better for the measurement; shadow or sunlight? What would be the consequences for long-term observations?

**AC1.7**: Thank you for the sensible comment. In Figure 9 there are two radiometric calibration panels. Where possible, we selected the panel under the best light condition. We didn't use the shaded areas, since the shadow partially hid the information about how the light was really reflected by the water and the suspended particles. This is a technical note illustrating the potentialities of the procedure and the criticalities that may be encountered during its practical applications. Our works in other case studies, under different hydrological and environmental conditions, aim to provide the best approach for long-term observations, among the possible concrete choices. It's not a simple answer.

**RC1.8**: The processing of the pixel values by subtracting the clear water image is very interesting and valid. I assume the improved relationship to the measured turbidity is due to the removal of the "noisy" background values. I would suggest to further focus on this analysis as it seems to be a promising step.

**AC1.8**: Thank you for the comment. This is a critical task currently under investigation; however, it is not the focus of the present study, which aims to highlight the preliminary outcomes of this pioneering experiment.

**RC1.9**: I would remove the UAV analysis due to a partly failed experiment (loss of information and faulty set camera parameters). The provided information might not be solid enough and need more experiments to being able to have reliable findings.

**AC1.9**: We decided to not remove the UAV results to show the outcome from a different camera and field view. Nevertheless, as explained in line 272, we removed the UAV results for the finale sensitivity analysis in the Discussion section, agreeing with your point of view.

**RC1.10**: Please, check the English language as there seem to be quite a few strange wordings.

**AC1.10**: We will review it.

**RC1.11**: Further comments

**AC1.11**: We agree to implement all these suggestions in the revised version of the manuscript.

**Author's Response to Reviewer 2**

**RC2.1**: You should mention more numerical results in your abstract. The Introduction section is quite dense with information. So it makes harder to the reader to get the key points. Consider simplifying the language and structure to improve readability.

**AC2.1**: We will review it, in order to make reading easier.

**RC2.2**: Please ensure that the main objectives and contributions of the study are clearly stated

**AC2.2**: Our objectives of developing and testing an image-based procedure for river turbidity estimation using camera systems within real environments, are already stated in the Abstract, in Section 1.2, and described in detail in Section 2. We'll try to emphasize them. The contributions of our study are highlighted in the Discussion Section 4.2, when we explained that the added value of our study is the direct application of the procedure in real conditions to understand its potential and the limitations, by critically analyzing the first results of the experiment shown.

**RC2.3**: Consider adding more detailed literature reviews that could strengthen the justification for the study.

**AC2.3**: For the same reason as the previous answer, there aren't a lot of significative and specific examples in literature on this. The proposed work wants to be a novel and provide added value in this research field, respect to the existing studies under laboratory-controlled conditions or under satellite spatial and temporal limitations in resolution, already cited in Section 1.2.

**RC2.4**: Clearly outline what specific research questions the study aims to answer and how the findings will contribute to existing knowledge in the field.

**AC2.4**: In our opinion, the main research question of this study is to find a continuous image-based procedure for river turbidity assessment, to understand the potential practical applications of different cameras and to identify an optimization of their installation setups, in order to offer new options for monitoring strategies and support existing techniques in deepening our knowledge of hydrological processes and aquatic ecosystems in rivers.

**RC2.5**: In the Methodology section, consider adding a brief summary at the beginning of the section that outlines how each method contributes to the overall goal of estimating turbidity.

**AC2.5**: In Section 2.2 there is already a brief summary of the overall image processing procedure and its importance for the turbidity estimation. Then, in the following subsection, the importance of each step of the procedure is highlighted. If needed, we'll refine Section 2, emphasizing it.

**RC2.6**: Ensure consistency in the use of terminology throughout the section.

**AC2.6**: we'll review it.

**RC2.7**: Discuss any limitations or uncertainties in the results.

**AC2.7**: We already answered this in the third and fifth question of the first reviewer. There, you will find more figures and considerations about accuracy and precision of each camera and band, properly compared with the measurements. We will consider adding them in the Discussion section.

**RC2.8**: Consider discussing the implications of the findings in terms of monitoring. Summarize the key findings and their implications in a concise manner. In conclusion, provide a brief overview of the main objectives and methodology of the study before diving into the conclusions.

**AC2.8**: In Discussion section, there is already a dedicated subsection called "Implications in river monitoring practices". If needed, we'll expand it with more considerations.

**RC2.9**: Based on the findings, suggest recommendations for future research or practical applications.

**AC2.9**: The purposes of this study were the description, and the field testing of our simplified image processing procedure tailored for water turbidity estimation in real riverine environments. We tested a wide range of camera types and installations setups, to explore their potential in actual river monitoring practices. On the light of these first results, the importance of conducting further experiments is clear.
In fact, we are currently finalizing the results of a second experiment, where the procedure has been tested under different hydrological conditions, using different cameras and tracers with different characteristics of the suspended particles. This will help us to generalize the method as much as possible. We'll clarify it better in the text.